# Design, Synthesis, Bioactivity Evaluation, Crystal Structures, and In Silico Studies of New α-Amino Amide Derivatives as Potential Histone Deacetylase 6 Inhibitors

**DOI:** 10.3390/molecules27103335

**Published:** 2022-05-22

**Authors:** Yangrong Xu, Hangjun Tang, Yijie Xu, Jialin Guo, Xu Zhao, Qingguo Meng, Junhai Xiao

**Affiliations:** 1National Engineering Research Center for the Emergency Drug, State Key Laboratory of Toxicology and Medical Countermeasures, Beijing Institute of Pharmacology and Toxicology, Beijing 100850, China; yrxuytu@163.com (Y.X.); beilou2353@163.com (Y.X.); shandongguojialin@163.com (J.G.); 2School of Pharmacy, Key Laboratory of Molecular Pharmacology and Drug Evaluation, Ministry of Education, Collaborative Innovation Center of Advanced Drug Delivery System and Biotech Drugs in Universities of Shandong, Yantai University, Yantai 264005, China; t1531994051@126.com; 3Department of Hepatology, Fifth Medical Center of Chinese PLA General Hospital, Beijing 100039, China; xuzhao080@outlook.com; 4China Military Institute of Chinese Materia, Fifth Medical Center of Chinese PLA General Hospital, Beijing 100039, China

**Keywords:** HDAC6 inhibitors, non-hydroxamate, α-amino amide, synthesis, bioactivity evaluation, crystal structure, reverse docking, molecular dynamics simulation

## Abstract

Hydroxamate, as a zinc-binding group (ZBG), prevails in the design of histone deacetylase 6(HDAC6) inhibitors due to its remarkable zinc-chelating capability. However, hydroxamate-associated genotoxicity and mutagenicity have limited the widespread application of corresponding HDAC6 inhibitors in the treatment of human diseases. To avoid such side effects, researchers are searching for novel ZBGs that may be used for the synthesis of HDAC6 inhibitors. In this study, a series of stereoisomeric compounds were designed and synthesized to discover non-hydroxamate HDAC6 inhibitors using α-amino amide as zinc-ion-chelating groups, along with a pair of enantiomeric isomers with inverted L-shaped vertical structure as cap structures. The anti-proliferative activities were determined against HL-60, Hela, and RPMI 8226 cells, and **7a** and its stereoisomer **13a** exhibited excellent activities against Hela cells with IC_50_ = 0.31 µM and IC_50_ = 5.19 µM, respectively. Interestingly, there is a significant difference between the two stereoisomers. Moreover, an evaluation of cytotoxicity toward human normal liver cells HL-7702 indicated its safety for normal cells. X-ray single crystal diffraction was employed to increase insights into molecule structure and activities. It was found that the carbonyl of the amide bond is on the different side from the amino and pyridine nitrogen atoms. To identify possible protein targets to clarify the mechanism of action and biological activity of **7a**, a small-scale virtual screen using reverse docking for HDAC isoforms (1–10) was performed and the results showed that HDAC6 was the best receptor for **7a**, suggesting that HDAC6 may be a potential target for **7a**. The interaction pattern analysis showed that the α-amino amide moiety of **7a** coordinated with the zinc ion of HDAC6 in a bidentate chelate manner, which is similar to the chelation pattern of hydroxamic acid. Finally, the molecular dynamics simulation approaches were used to assess the docked complex’s conformational stability. In this work, we identified **7a** as a potential HDAC6 inhibitor and provide some references for the discovery of non-hydroxamic acid HDAC6 inhibitors.

## 1. Introduction

Histone acetylation and deacetylation are very important epigenetic mechanisms regulating gene expression [1]. The acetylation level of the lysine residues on histone tails is regulated by histone acetyltransferases (HATs) and histone deacetylases (HDACs) [2]. HATs transfer acetyl groups to *ε*-lysine residues in histones, resulting in a reduction in the positive charge of the histone tail, which weakens the interaction with a negatively charged DNA backbone, leaving chromatin in a relaxed state. This relaxed state results in increased accessibility of transcription factors to DNA, which in turn activates transcription of the associated genes. In contrast, HDACs compact chromatins and silence associated genes [3]. In addition, lysine acetylation can also occur in non-histone proteins, suggesting that HATs and HDACs are multifunctional factors that not only act on transcription but also in various other cellular processes [4]. It have been reported that the dysregulation of HDACs is implicated in many diseases, such as cancer [5], autoimmune [6], and psychiatric diseases [7]. Consequently, HATs and HDACs have emerged as promising targets for small-molecule drug discovery [8]. At the present time, four HDACs inhibitors (Figure 1a), namely vorinostat (SAHA), belinostat (PXD-101), panobinostat (LBH-589), and romidepsin (FK228), have been approved for the treatment of refractory or relapsed cutaneous, peripheral T cell lymphomas, or multiple myeloma [9]. Moreover, chidamide was another potent HDAC inhibitor developed and approved in China for the treatment of peripheral T cell lymphomas [10].

To date, eighteen HDAC isozymes have been identified and classified in mammals, based on sequence homology to yeast protein orthologues, into four classes: class I (HDAC1, 2, 3, 8), class IIa (HDAC4, 5, 7, 9), class IIb (HDAC 6, 10), and class IV (sole HDAC11) are Zn^2+^-dependent enzymes, whereas class III HDACs (sirtuins 1–7) are NAD^+^-dependent enzymes [11]. HDAC6 is the only isoenzyme of the HDAC family with two functional active sites, CD1 and CD2. The key catalytic steps of the deacetylation reaction revealed that CD2 has broad substrate specificity. However, CD1 is highly specific for the hydrolysis of C-terminal of acetyl-lysine residues [12]. Unlike other HDACs, which predominantly exist in the nucleus, HDAC6 mainly locates in the cytoplasm and acts as a workhorse for regulating the acetylation status of non-histone substrates, including α-tubulin [13], cortactin [14], and heat shock protein 90 (HSP90) [15]. Therefore, HDAC6 plays specific physiological roles in multiple cellular pathways, including cell movement, endocytosis, cell autophagy, apoptosis, and protein transport and degradation [16,17]. Another unique feature of HDAC6 is that no apparent deficient phenotypes or lethal effects were observed in HDAC6 knockout mice [18,19,20], suggesting that HDAC6 is an ideal target for antitumor drug discovery and development. In 2016, Yang Hai unveiled the crystal structures of HDAC6 [12], which promoted the investigation of HDAC6 inhibitors. Since then, hundreds of HDAC6 inhibitors have been discovered or are being discovered.

Hydroxamate, as a zinc-binding group (ZBG), prevails in the design of HDAC6 inhibitors due to its remarkable zinc-chelating capability. For instance, tubacin (Figure 1b) [21], citarinostat (ACY-241) [22], ACY-738 [23], tubastatin A [24], and ricolinostat (ACY-1215) [25] are known as representative HDAC6 inhibitors characterized by its ZBG being hydroxamic acid. However, hydroxamate-based inhibitors suffer from some undesirable drawbacks that often lead to clinical discontinuation, such as mutagenicity [26], poor pharmacokinetic properties and severe side effects [27,28,29]. Therefore, medicinal chemists began to realize that the hydroxamate group is not necessarily the best ZBG for the drug discovery of HDAC6 inhibitors and shifted their interest to the non-hydroxamate structure. In recent years, few novel hydroxamate alternatives have been discovered, such as mercaptoacetamide, 3,3,3-trifluorolactic amide, difluoromethyl, and trifluoromethyl substituted 1,3,4-oxadiazole. Compared with hydroxamate, several representative non-hydroxamate HDAC6 inhibitors a-f (Figure 1c) showed good pharmacological effects and druglike properties and were unlikely to cause mutagenicity [30,31,32,33]. To enrich the structural types of non-hydroxamic acid HDAC6 inhibitors, a series of compounds were designed by introducing α-amino amide as ZBG and synthesize. Their anti-proliferative activities and cytotoxicity were evaluated. Their structure–activity relationship was preliminarily analyzed. To better understand the relationship between molecule structure and activity, we carried out the studies of X-ray single crystal diffraction. 

Virtual screening has become an important part of the drug discovery process. Molecular docking-based virtual screening involves docking and scoring a library of small-molecule compounds against a given target protein, from which potential ligand molecules for the target protein are screened. Reverse molecular docking is the opposite of molecular docking in that it involves docking and scoring an active molecule with multiple or a large number of protein targets, thus predicting the potential targets of the active molecule. This strategy can help to clarify the mechanism of action and biological activities of compound whose target is ambiguous [34,35,36]. So far, this approach has been used in many studies as a primary or secondary option for identifying small-molecule targets [37,38,39]. Based on this, we performed reverse docking to predict the target of active molecules. Since these compounds are designed for HDAC6, we assume that HDAC6 will be the best receptor for the active molecule among the HDAC isomers. To confirm this, we performed a small-scale reverse docking based on virtual screening where the active molecule was docked with ten HDAC isoforms (1–10). The interaction patterns were analyzed using molecular docking, and molecular dynamics simulation approaches were carried out to assess the docked complex’s conformational stability. This work is expected to provide some implications for the rational design of non-hydroxamate HDAC6 inhibitors.

## 2. Results and Discussion

### 2.1. Design

We analyzed the crystal structure of (*R*)-TSA-human HDAC6 complex (PDB ID: 5EDU) for compound design. The HDAC6 active site is a narrow hydrophobic cavity with a length of about 11 Å (Figure 2a). Zinc ion is at the bottom of the cavity, off to one side. The hydrophobic pocket is narrow, with a diameter of about 3.14 Å (Figure 2b). Structures with a diameter larger than the benzene ring may be blocked out of the cavity. The surface of the cavity is a huge rectangular groove, which is almost perpendicular to the cavity. The rim of the cavity is mostly hydrophobic, while the distal part is hydrophilic. The crystal structure of (*R*)-TSA-HDAC6 complex shows that TYP782, PHE620, PHE680, SER568, HIS611, HIS610, and ASP649 are important amino acid residues for site formation (Figure 2c).

The typical pharmacophore model of HDAC6 inhibitors consists of three regions: cap, linker, and a zinc-binding group (ZBG). We designed the structure accordingly for different parts (Figure 3). In the zinc-binding group region, we introduced α-amino amide to chelate with zinc ion due to its similar chelation mode with hydroxamic acid and mercaptoacetamide. Moreover, the methyl group was introduced at the carbonyl α-position of the α-amino amide to increase the chirality and the structural diversity of α-amino amide. Biphenyl was used as a linker because of its larger hydrophobicity and the ability to form π–π interactions with PHE620 and PHE680. The pyridine nitrogen atom may chelate zinc ion when it is on the same side as the carbonyl oxygen atom. To increase the chance of chelation with zinc ion, we replaced the benzene ring with the pyridine ring according to bioisosterism. It also increases the possibility of hydrogen bond formation between the pyridine nitrogen atom and the phenolic hydroxyl oxygen atom of TYP782. It is well known that chirality has an important influence on the selectivity and activity of compounds. It has been reported that *R*-stereoisomer of TSA (Figure 3) is found to be strong dual inhibitor of both HDAC6 and HDAC1, while the unnatural enantiomer (*S*)-TSA is found to be extremely selective to HDAC6 [34]. Inspired by the different selectivity of (*R*)-TSA and(*S*)-TSA enantiomers for HDAC6, we designed a pair of stereoisomers 1-[(4-bromophenyl) sulfonyl]-L-proline methyl ester and1-[(4-bromophenyl) sulfonyl]-D-proline methyl ester as the cap recognition region scaffold to investigate the influence of chirality on activity. The crystal structures show that they have an inverted L-shaped vertical structure and are stereoisomers of each other. The pyrrolidine ring can interact with the hydrophobic rim of the cavity, while the benzene ring can extend into the cavity. Therefore, this pair of scaffolds may have a perfect fit with the active pocket of HDAC6 and there may be differences in activity.

### 2.2. Synthesis

As shown in Figure 1, target compounds **7a**–**f** and **13a**–**f** were prepared in seven steps, including the Hinsberg reaction, the esterification reaction, the Miyaura borylation reaction, the Suzuki–Miyaura cross-coupling reaction, the nitro reduction reaction, the amide bond condensation reaction, and the Fmoc removal reaction. First, according to reported literature method [40], the reaction of *p*-benzenesulfonyl chloride with D-proline and L-proline obtained intermediates **1a**(*R*) and **1b**(*S*), respectively, which were directly used in the next reaction step. Subsequently, a pair of stereoisomers (*R*)-1-[(4-bromophenyl) sulfonyl]-D-proline methyl ester **2** and (*S*)-1-[(4-bromophenyl) sulfonyl]-L-proline methyl ester **8** were obtained by the esterification reaction, using SOCl_2_ as the acylation reagent in MeOH. Next, the reactions of intermediates **2** and **8** with bis(pinacolato)diboron in the presence of Pd(dppf)Cl_2_ and potassium acetate at 90 °C for 3 h afforded intermediates **3** and **9**. Intermediates **3** and **9** then reacted with 1-bromo-4-nitrobenzene or 5-bromo-2-nitropyridine via the Suzuki cross-coupling reaction to give **4a**, **4b**, **10a**, and **10b**. Afterwards, **5a**, **5b**, **11a**, and **11b** were prepared by reducing the nitro group to the amino group with zinc powder in the presence of AcOH. Then, **6a**–**f** and **12a**–**f** were subjected to the amide bond-forming reaction with Fmoc-glycine, Fmoc-L-alanine, and Fmoc-D-alanine in a method which featured EDCI/HOBt/DIPEA as a coupling combination. Finally, target compounds **7a**–**f** and **13a**–**f** were obtained by reacting **6a**–**f** and **12a**–**f** with 20% piperidine in DMF for 30 min.

### 2.3. Anti-Proliferative Activity

To explore the anti-proliferative activity of the synthesized compounds, we chose HL-60 [41,42], Hela [43,44], and RPMI 8226 [45,46] cancer cell lines to perform the assay according to reported literatures related to HDAC6. As results summarized in Table 1, at 50 μM, **7a**, **13a**, **13b**, **13c**, **13d**, **13e**, and **13f** showed remarkable inhibition against all three tumor cell lines. At 10 μM, **7a** and **13a** inhibited tumor cells significantly more than other compounds, especially when against Hela tumor cell line. Therefore, we further tested the IC_50_ values of **7a**, **13a**, **13b**, **13c**, and **13d** against HL-60, Hela, and RPMI 8226 cancer cell lines (Table 2). Notably **7a** and its stereoisomer **13a** exhibited excellent activities against Hela cells with IC_50_ = 0.31 µM and IC_50_ = 5.19, respectively. As we expected, there is a clear difference between the two stereoisomers. Further evaluation of cytotoxicity toward human normal liver cells HL-7702 was carried out. As summarized in Table 3, the 50% cytotoxic concentration (CC_50_) of **7a** against human normal liver cells HL-7702 is 21.07 μM, and the selection index SI is 67.97, indicating its safety for normal cells. From the activity results, we can draw a preliminary structure–activity relationship, although more compounds are active when the cap region of the compound is L-proline than when the cap region is D-proline, as the activity is stronger when the cap region is D-proline. Even more puzzling, the activity of the compounds is lost or significantly weakened when only one carbon atom of benzene ring is changed to nitrogen atom, or when methyl is introduced at the α position of the carbonyl group of the amide bond.

### 2.4. Crystal Structures

To clearly and distinctly understand the structure of **2**, **8**, **7a**, **7d**, and **13a**, we cultured their crystals. Suitable single crystals of **2** and **8** were obtained by slow evaporation of a solution of the compounds in ethyl acetate at room temperature. Suitable single crystals of **7a**, **7d**, and **13a** were obtained by slow evaporation of a solution of the compounds in methanol at room temperature. Crystal data, data collection, and structure refinement details for **2**, **8**, **7a**, **7d**, and **13a** are summarized in Table 4.

As shown in Figure 4, the asymmetric unit of **2** and **8** contains one independent molecule. The bond angles of N11-S7-C1 in compounds **2** and **8** are 107.87(9)° and 107.72(5)°, respectively. The C16-C12-N11-S7 torsion angle of **2** is 97.42(7)°, whereas the corresponding torsion angle of **8** is −96.42(1), indicating a different conformation at the C12 position. The mean plane of the phenyl ring, defined as C1-C2-C3-C4-C5-C6, and pyrrolidine ring, defined as N11–C12–C13–C14–C15, formed angles of 82.82(7)° in **2**, revealing that the phenyl and pyrrolidine rings are almost perpendicular. Methyl carboxylate extends upwards instead of downwards. If compound **2** is rotated by 180°, it can be imagined that the position of methyl carboxylate is at the position of C14 of compound **8**. They are mirror images of each other, but they cannot completely overlap. This steric differences between **2** and **8** may be the reason for the different activities of **7a** and **7d.**

As shown in Figure 5, the structures of compounds **7a**, **7d**, and **13a** all contain two molecules in their unit cells, and the difference between the two molecules is minimal. Compounds **7a** and **13a** are almost identical except that they are stereoisomeric. The two molecules in the **7a** unit cell interact through intermolecular hydrogen bonds O10…H3-N3 and non-classical hydrogen bonds O9…H6-C6. The distances from C7 to C20 and C19 in **7a** are 10.90(5) Å and 9.5328 Å, respectively. It is worth emphasizing that, from the crystal structure of **7a**, **7d**, and **13a**, we found that the carbonyl oxygen atom of the amide bond is in a trans configuration with the amino group, indicating that this configuration is a stable configuration. It is worth mentioning that **7d** may be affected by the pyridine nitrogen atom, and the two molecules in the unit cell are head to tail, while the two molecules in **7a** and **13a** unit cells are head to head. The two molecules in the unit cell of **7d** form a hydrogen bond O9…H-N4 and O3…H-N8 due to the head-to-tail connection. The biphenyl plane in **7a** and **13a** is nearly perpendicular to the biphenyl plane of the other molecule. However, in the **7d** unit cell, the planes of the two phenylpyridine rings are parallel. The distances from C7 to C19 and C20 in **7a** are 9.5328 Å and 10.90(5) Å, respectively.

### 2.5. In Silico Studies

#### 2.5.1. Inverse Docking

To identify a possible protein target to clarify the mechanism of action and biological activity of **7a**, we explored reverse docking as a way to implement small-scale target fishing for the **7a**. The reverse docking studies were carried out in four steps: method development, method validation, virtual screening, and interaction pattern analysis.

Firstly, we used the HDAC6 protein (5edu) and the co-crystallised compound trichostatin A (TSN) as subjects for method development. The co-crystallized ligand TSN was docked into HDAC6 using Discovery Studio 4.5 and Schrödinger 2021, but the bidentate chelation pattern was not obtained by using the former docking program, so the schrödinger program was used for in silico studies reported here. In addition, we used the standard precision (SP) mode and the extra precision (XP) mode to conduct the docking experiments, respectively, and found that the SP mode could effectively obtain the bidentate chelation mode, but the XP mode could not. In 2021, Kashyap et al. performed molecular docking with HDAC6 using the Glide XP mode and found that the majority hydroxamic acid inhibitors coordinated with the zinc ion in monodentate chelation, which is consistent with our docking result in the same model [47]. Therefore, we chose to use the SP mode for in silico studies. Subsequently, we established a standard docking operation procedure in the SP mode, and using this docking method, 9 out of 10 docking results formed a bidentate chelate pattern.

To evaluate the accuracy and precision of the established docking protocols, self-docking was performed in the SP mode of glide. The co-crystallised ligands of HDAC isoforms (1, 2, 4, 6, 7, 8, 10) were extracted and subsequently docked to their corresponding protein targets. Two-dimensional (2D) interaction pattern diagrams for self-docking are presented in the Appendix A. The self-docking results showed that HDAC2, HDAC6, and HDAC8 formed a bidentate chelation pattern while HDAC1, HDAC4 HDAC7, and HDAC10 formed a monodentate chelation pattern with their corresponding ligand, which were consistent with their original crystal complexes. HDAC3, HDAC5, and HDAC9 were not subjected to self-docking because they did not have a suitable ligand. Root mean square deviation (RMSD) values of self-docked poses, with respect to the co-crystallized ligand conformation, were computed. An RMSD value of less than 2 Å is indicative of a good docking methodology. The self-docked poses of all ligands showed good overlap with the co-crystallised ligand orientations (Figure 6) and the evaluated RMSD values are listed in Table 5. All the RMSD values are less than 2 Å except HDAC1, whose co-crystallized ligand is a bulky peptide.

After validation, a docking-based reverse virtual screening of **7a** and TSN against HDAC isoforms 1–10 was carried out. Compound **7a** and TSN were individually docked with each HDAC isoform. Each docking score was calculated. The basic principle of reverse docking is that the binding strength of a small-molecule ligand and a potential protein target is determined by their interaction energy (docking energy). Generally, a more negative docking energy indicates a stronger bond between the ligand and the receptor (protein target). Furthermore, the receptor is more likely to be the target of the query molecule. The reverse docking scores of **7a** and TSN against HDAC isoforms 1–10 were listed in a table. It can be seen from the values that **7a** has the lowest docking score with HDAC6 (−8.57 kcal/mol). At the same time, the docking score of TSN for each isomer is generally lower than that of **7a**. Through virtual screening, we concluded that HDAC6 is a potential target of **7a**; in other words, **7a** has a certain selectivity for HDAC6 over other isoforms.

To explore the plausibility of the predicted results, we analyzed the interaction mode of **7a** with each HDAC isoform. By analyzing the output docking results, we found that **7a** forms a bidentate chelation mode with HDAC2, HDAC6, and HDAC10. Moreover, HDAC6 forms the largest numbers of the bidentate chelation mode compared to other HDAC isomers with **7a**. As we all know, the coordination interaction of the inhibitors with the catalytic Zn^2+^ is essential for HDAC inhibition. The bidentate chelation can better compete for Zn^2+^ ion compared to the monodentate chelation, and many reported studies have been devoted to the discovery of bidentate chelation inhibitors. Therefore, we consider our virtual screening results to be reasonable in terms of coordination patterns. A 2D interaction map of the best docking results of **7a** with each HDAC isoform is presented in the Appendix A.

After identifying HDAC6 as the target of **7a**, we carried out forward docking of **7a**, **7d**, and **13a** to try and explore the reasons for their differences in activity, while TSN was used as the reference molecule. Structures of **7a**, **7d**, **13a**, and TSN were optimized and docked with HDAC6 at the same time. Their 2D and 3D interaction diagrams are shown in Figure 7 and Figure 8, respectively. The α-amino amide moiety of **7a** coordinates to Zn^2+^ in a bidentate fashion, forming a five-membered chelate complex with Zn^2+^-O distances of 2.2 Å and 2.5 Å for the −NH_2_ and C=O groups, respectively (Figure 7a). Compound **7a** also forms hydrogen bonds with HIS 610, TYR782, and GLY619. Three π–π stacking interactions with PHE620, PHE680, and HIS651 are formed through the phenyl linker group. The binding pattern of **7a** in the active pocket of HDAC6 is similar to that of TSN (Figure 7d). TSN forms bidentate coordination fashion through its hydroxamic group. TSN forms two hydrogen bonds with HIS 610 and TYR782 and one π–π stacking interaction with PHE 620. The docking patterns of **7d** (Figure 7b) and **13a** (Figure 7c) with HDAC6 are similar, and the α-amino amide moiety of **7a** coordinates to Zn^2+^ in a monodentate chelate manner. Compounds **7d** and **13a** form three hydrogen bonds with HIS 610, PHE680, and GLY619, and three π–π stacking interactions with PHE620, PHE680, and HIS651 According to the above docking results, we believe that the main reason for the difference in activity between **7a** and other molecules is that **7a** can better form a bidentate coordination with Zn^2+^, which is a key factor that has been recognized to inhibit HDAC6 protein.

#### 2.5.2. Molecular Dynamic Simulation

To have more insights about the binding recognition in a solvated and all-atom flexible environment, **7a** with the best SP docking score (−8.57 kcal/mol) was subjected to molecular dynamics (MD) simulations for 100 ns with HDAC6. Overall, 1000 frames were generated in the trajectory. Protein–ligand interaction stability throughout the simulation was studied using root mean square deviation (RMSD) analysis.

As shown in Figure 9A, **7a** and HDAC6 are largely paced and maintain a steady state after 30 ns, indicating that **7a** is stable with respect to the protein and its binding pocket. Figure 9B demonstrates the conformational changes taking place along the HDAC6 protein side chain. Protein residues that interact with the ligand are marked with green-colored vertical bars. Root mean square fluctuation (RMSF) data of the protein depict the flexibility from 0.80 to 7.4 Å.

Protein interactions with the ligand are monitored throughout the simulation (Figure 9C).

These interactions are divided into four main categories, including hydrogen bonds(H-bonds), hydrophobic interactions, ionic bonds, and water bridges Hydrogen bonds are formed with HIS 610, HIS 611, HIS 619, GLU 779, and TYR 782. Of these, hydrogen bonds formed by HIS 610, HIS 619, and TYR 782 are maintained during more than 30% simulation time. Hydrophobic interactions are formed with PHE 620, PHE 680, and TYR 782. Among of these, the hydrophobic interactions with PHE 620 and PHE 680 are maintained during more than 50% of the simulation time. Ionic bonds are formed with ASP 649, HIS 651, and ASP742. Among them, ASP 649 forms two ionic bonds with zinc ions. Water bridges are formed with ASP 567 and GLU 779.

A schematic diagram of the interaction of **7a** with protein residues is shown in Figure 9D. Interactions that occur with more than 30.0% of the simulation time in the selected trajectory are shown. From the picture, we can see that **7a** exhibits bidentate chelation to the Zn^2+^ ion, which is maintained throughout the simulation. In addition, hydrogen bonding formed with HIS 610, GLY 19, and TYR 782 may also play an important role in stabilizing the complex.

## 3. Discussion

In this paper, a series of stereoisomeric α-amino amide-based non-hydroxamate HDAC6 inhibitors were designed and synthesized, and compound **7a** demonstrated excellent anti-proliferative activities. The structure–activity relationships of these compounds have been preliminarily analyzed. Stereoscopic differences in the structure of the cap region can lead to differences in activity. The activities of the compounds are lost or significantly weakened when only one carbon atom of benzene ring is changed to the nitrogen atom, or when methyl is introduced at the α position of carbonyl group of amide bond. To better understand the relationship between molecule structure and activity, we carried out X-ray single crystal diffraction. We predicted the target of **7a** by reverse docking and concluded that HDAC6 may be its potential target. The α-aminoamide moiety of **7a** forms a bidentate chelate conformation with the zinc ion in the active pocket of HDAC6. We consider that **7a** may inhibit HDAC6 through this interaction mode, thereby playing its anti-tumor effect against Hela cancer cells. It can be seen from the docking mode of **7a** with HDAC6 that our designed compound still has some shortcomings, but the 2-amino-*N*-phenylacetamide moiety has high ligand efficiency; thus, this structural fragment can be used to carry out structural modification. Moreover, the molecular docking study provides us with the judgment of the compound structure and the position of the zinc ion and provides certain guidance for the subsequent structural modification. Molecular dynamics simulation approaches were used to assess the docked complex’s conformational stability. Coordination interactions with zinc ions and hydrogen bonding formed with HIS 610, GLY 19, and TYR 782 may play an important role in stabilizing the complex. The structure of **7a** can be optimized by introducing amino, oxime, hydroxyl, trifluoromethyl, sulfhydryl, and other structures to the α position of the amide bond to enhance inhibitor activity. Stereoisomeric scaffolds **2** and **8** can also be used as a tool in the discovery of inhibitors for different targets. Further structural modification studies of **7a** are underway.

## 4. Materials and Methods

### 4.1. Chemical Reagents and Instruments

Reagents and solvents were purchased from commercial suppliers and used without further purification. All the reactions were monitored by TLC using silica gel TLC plates (GF254). Silica gel (200–300 mesh) was used for chromatography. Melting points were determined on a Buchi melting point apparatus (M-565). The ^1^H and ^13^C NMR spectra were recorded on a JEOLECA400 spectrometer, with TMS as an internal standard at ambient temperature. All chemical shifts are reported in parts per million (ppm). All coupling constants were reported in Hertz. The HRMS was recorded on an Agilent TOF G6230A mass spectrometer.

### 4.2. The Synthesis of Compounds **7a**–**f**, **13a**–**f**

#### 4.2.1. (2*R*)-1-(4-Bromobenzenesulfonyl)pyrrolidine-2-carboxylic Acid (**1a**)

D-proline (8.06 g, 70 mmol) was added to water (168 mL), and stirred at room temperature until the D-proline dissolved. *p*-Bromobenzenesulfonyl chloride (21.46 g, 84 mmol) was added to the above mixture at 0 °C, followed by sodium carbonate (8.90 g, 84 mmol). After 10 min, the reaction solution was transferred to room temperature and reacted overnight. The pH value of reaction mixture was adjusted to >9 with sodium hydroxide. The mixture was washed with ethyl acetate (50 mL), and the pH value of aqueous phase was adjusted to <3 with concentrated hydrochloric acid. The acidic aqueous phase was extracted with ethyl acetate (300 mL), and the organic phase was washed with saturated brine solution, dried over anhydrous sodium sulfate overnight, and filtered. The filtrate was concentrated to dryness in vacuo to give **1a** (18.14 g, 84.5% yield) as an oil.

#### 4.2.2. (2*R*)-1-(4-Bromobenzenesulfonyl)pyrrolidine-2-carboxylate Methyl Ester (**2**)

Thionyl chloride (9.85 mL, 135.7 mmol) was added dropwise to a solution of **1a** (18.14 g, 54.28 mmol) in anhydrous methanol (110 mL) at an ice bath. The mixture was stirred for 2 h at reflux and then concentrated in vacuo. The residue was diluted with ethyl acetate (300 mL), then washed with deionized water and saturated saline solution, dried over anhydrous sodium sulfate overnight, and filtered. The filtrate was concentrated to dryness in vacuo to give **2** (19.04 g, 92.1% yield).

#### 4.2.3. (2*R*)-1-[4-(4-Methyl-1,3,2-dioxaborolane-2-yl)benzenesulfonyl]pyrrolidine-2-carboxylate Methyl Ester (**3**)

At an N_2_ atmosphere, **2** (19.04 g, 50 mmol), [1,1′-Bis(diphenylphosphino)ferrocene]dichloropalladium(II) (1.83 g, 2.5 mmol), potassium acetate (14.72 g, 150 mmol), and bispinacol boronate (15.24 g, 60 mmol) were suspended in anhydrous DMF (100 mL) and stirred for 3 h at 90 °C. The mixture was concentrated in vacuo. The residue was diluted with ethyl acetate and filtered through celite. The filtrate was washed with deionized water and saturated saline solution successively, dried over anhydrous sodium sulfate overnight, and filtered. The filtrate was concentrated to dryness in vacuo and purified on a silica gel column to give **3** (16.57g, 83.9% yield) as a white solid.

#### 4.2.4. (2*R*)-1-[4-(4-Nitrophenyl)benzenesulfonyl]pyrrolidine-2-carboxylic Acid Methyl Ester (**4a**)

At an N_2_ atmosphere, compound **3** (1.80 g, 4.56 mmol), 1-iodo-4-nitrobenzene (2.27 g, 9.12 mmol), tetrakis(triphenylphosphine)palladium (0) (92.44 mg and 0.08 mmol), and anhydrous sodium carbonate (1.06 g, 10 mmol) were suspended in DMF/H_2_O (13 mL/3 mL) and stirred for 3 h at 90 °C. The mixture was concentrated in vacuo, diluted with ethyl acetate, and filtered through celite. The filtrate was washed with deionized water and saturated saline solution successively, dried over anhydrous sodium sulfate overnight, and filtered. The filtrate was concentrated to dryness in vacuo and purified on a silica gel column to give **4a** (1.06 g, 59.6% yield) as a yellow solid. m.p. 165.2~168.0 °C; ^1^H NMR (400 MHz, DMSO-*d_6_*) δ 8.35 (d, *J* = 9.0 Hz, 2H), 8.08–8.01 (m, 4H), 7.97 (d, *J* = 8.6 Hz, 2H), 4.30 (dd, *J* = 8.7, 4.0 Hz, 1H), 3.66 (s, 3H), 3.42 (ddd, *J* = 9.7, 7.2, 5.0 Hz, 1H), 3.24 (dt, *J* = 9.8, 7.1 Hz, 1H), 2.06–1.94 (m, 1H), 1.94–1.79 (m, 2H), 1.72–1.61 (m, 1H). ^13^C NMR (100 MHz, DMSO-*d*_6_) δ 172.14, 147.42, 144.71, 142.16, 137.57, 128.55, 128.38, 128.04, 124.27, 60.29, 51.80, 48.49, 31.05, 24.91. HR-MS(TOF): calcd. for C_18_H_18_N_2_SO_6_, [M + H]^+^: 391.0964, found: 391.0958.

Compounds **4b**, **10a**, and **10b** were obtained using the synthesis method of **4a**.

#### 4.2.5. (2*R*)-1-[4-(6-Nitropyridin-3-yl)phenyl]pyrrolidine-2-carboxylic Acid Methyl Ester (**4b**)

Yellow solid (1.19 g, yield 76.3%); m.p. 121.0~124.0 °C; ^1^H NMR (400 MHz, DMSO-*d*_6_) δ 9.09 (d, *J* = 2.3 Hz, 1H), 8.62 (dd, *J* = 8.5, 2.4 Hz, 1H), 8.44 (d, *J* = 8.5 Hz, 1H), 8.17–8.09 (m, 2H), 8.04–7.98 (m, 2H), 4.32 (dd, *J* = 8.6, 4.0 Hz, 1H), 3.66 (s, 3H), 3.43 (ddd, *J* = 9.6, 7.1, 4.9 Hz, 1H), 3.24 (dt, *J* = 9.8, 7.1 Hz, 1H), 2.07–1.95 (m, 1H), 1.95–1.79 (m, 2H), 1.67 (td, *J* = 7.0, 5.9, 3.8 Hz, 1H). ^13^C NMR (100 MHz, DMSO-*d*_6_) δ 172.15, 156.10, 147.40, 139.60, 139.33, 139.07, 138.12, 128.73, 128.10, 118.62, 60.32, 52.24, 48.49, 30.47, 23.79. HR-MS(TOF): calcd. for C_17_H_17_N_3_SO_6_, [M + H]^+^: 392.0916, found: 392.0911.

#### 4.2.6. (2*S*)-1-[4-(4-Nitrophenyl)benzenesulfonyl]pyrrolidine-2-carboxylate Methyl Ester (**10a**)

White solid (0.91g, yield 58.2%); m.p. 164.1~166.5 °C; ^1^H NMR (400 MHz, DMSO-*d_6_*) δ 8.35 (d, *J* = 2.0 Hz, 1H), 8.34 (d, *J* = 2.1 Hz, 1H), 8.07–8.01 (m, 4H), 7.99–7.95 (m, 2H), 4.30 (dd, *J* = 8.6, 4.1 Hz, 1H), 3.66 (s, 3H), 3.42 (ddd, *J* = 9.8, 7.2, 5.0 Hz, 1H), 3.23 (dt, *J* = 9.7, 7.1 Hz, 1H), 2.05–1.94 (m, 1H), 1.94–1.79 (m, 2H), 1.66 (dd, *J* = 7.0, 4.9 Hz, 1H). ^13^C NMR (100 MHz, DMSO-*d_6_*) δ 172.14, 147.41, 144.70, 142.15, 137.57, 128.25, 60.29, 52.21, 48.49, 30.45, 24.29. HR-MS (TOF): Calcd. For C_18_H_18_N_2_SO_6_, [M + H]^+^: 391.0964, found: 391.0958.

#### 4.2.7. (2*S*)-1-[4-(6-Nitropyridin-3-yl)phenyl]pyrrolidine-2-carboxylic Acid Methyl Ester (**10b**)

Yellow solid **10b** (1.23g, yield 78.4%); m.p. 121.2~123.8 °C; ^1^H NMR (400 MHz, DMSO-*d_6_*) δ 9.09 (dd, *J* = 2.5, 0.7 Hz, 1H), 8.62 (dd, *J* = 8.5, 2.4 Hz, 1H), 8.44 (dd, *J* = 8.6, 0.7 Hz, 1H), 8.12 (d, *J* = 8.7 Hz, 2H), 8.00 (d, *J* = 8.5 Hz, 2H), 4.32 (dd, *J* = 8.6, 4.0 Hz, 1H), 3.66 (s, 3H), 3.43 (ddd, *J* = 9.8, 7.2, 5.0 Hz, 1H), 3.24 (dt, *J* = 9.7, 7.1 Hz, 1H), 2.04–1.97 (m, 1H). 1.94–1.80 (m, 2H), 1.67 (dt, *J* = 7.1, 4.9 Hz, 1H). ^13^C NMR (100 MHz, DMSO-*d_6_*) δ 172.13, 156.08, 147.81, 139.58, 139.30, 139.05, 138.12, 128.71, 128.08, 118.59, 59.45, 52.94, 47.94, 30.45, 24.92. HR-MS(TOF): calcd. for C_17_H_17_N_3_SO_6_, [M + H]^+^: 392.0916, found: 392.0910.

#### 4.2.8. (2*R*)-1-[4-(4-Aminophenyl)benzenesulfonyl]pyrrolidine-2-carboxylic Acid Methyl Ester (**5a**)

At an N_2_ atmosphere, **4****a** (1.21 g, 3.10 mmol), zinc powder (810.96 mg and 12.4 mmol) and ammonium chloride (248.73 mg and 4.65 mmol) were suspended in water (19 mL) and stirred for 5 h at 80 °C. After the reaction was completed, ethyl acetate was added to the reaction system, stirred, and filtered through celite. The filtrate was washed with deionized water and saturated saline solution successively, dried over anhydrous sodium sulfate overnight, and filtered. The filtrate was concentrated to dryness in vacuo and purified on a silica gel column to give **5a** (0.61g, 54.5% yield) as a yellow solid.

According to the synthesis method, **5a**, **4b**, **10a**, and **10b** were reduced to obtain the corresponding **5b**, **11a**, and **11b**.

#### 4.2.9. General Method of Amide Condensation Reaction

To a 25 mL single-neck bottle, anhydrous DMF (5 mL), Fmoc-amino acid (1.7 mmol), EDCI (375.7 mg and 1.96 mmol), and HOBt (229.72 mg and 1.7 mmol) were added, respectively, followed by the Fmoc-amino substrate (0.85 mmol). The mixture was stirred for 2 h at room temperature. After the reaction was completed, the mixture was concentrated in vacuo, diluted with dichloromethane, and washed with saturated sodium carbonate solution. The organic phase was washed with deionized water and saturated saline solution successively, dried over anhydrous sodium sulfate, and filtered. The filtrate was concentrated to dryness in vacuo and purified on a silica gel column to give target compound as a light yellow oil.

Compounds **6a**–**c** were synthesized using the reaction of **5a** with Fmoc-glycine, Fmoc-D-alanine, and Fmoc-L-alanine. Compounds **6d**–**f** were synthesized by the reaction of **5b** with Fmoc-glycine, Fmoc-D-alanine, and Fmoc-L-alanine. Compounds **12a**–**c** were synthesized using the reaction of **11a** with Fmoc-glycine, Fmoc-D-alanine, and Fmoc-L-alanine. Compounds **12d**–**f** were synthesized by the reaction of **11b** with Fmoc-glycine, Fmoc-D-alanine, and Fmoc-L-alanine.

#### 4.2.10. Synthesis of Compounds **7a**–**7f** and **12**a–**12f**

To 12 reaction vials containing 0.5 mmol of **6a**–**f** and **11a**–**f**, respectively, DMF (2.5 mL) was added, followed by piperidine (92 μL, 1.0 mmol). The mixture was stirred for 30 min at room temperature. The mixture was concentrated in vacuo, diluted with dichloromethane. The organic phase was washed with deionized water and saturated saline solution successively, dried over anhydrous sodium sulfate overnight, and filtered. The filtrate was concentrated to dryness in vacuo and purified on a silica gel column to give target compounds.

#### 4.2.11. (2*R*)-Methyl 1-{4-[4-[4-(2-Aminoacetamido)phenyl]benzenesulfonyl}pyrrolidine-2-carboxylate (**7a**)

White solid (134 mg; yield 64.2%), m.p. 163.4~165.3 °C; ^1^H NMR (400 MHz, DMSO-*d_6_*) δ 7.89 (q, *J* = 8.4 Hz, 4H), 7.83–7.69 (m, 4H), 4.24 (d, *J* = 4.5 Hz, 1H), 3.66 (s, 3H), 3.31 (s, 2H), 3.21 (s, 2H), 1.98 (d, *J* = 6.2 Hz, 1H), 1.86 (d, *J* = 12.6 Hz, 2H), 1.63 (d, *J* = 6.4 Hz, 1H). ^13^C NMR (100 MHz, DMSO-*d*_6_) δ 172.37, 172.20, 144.08, 139.46, 135.42, 132.69, 127.90, 127.55, 126.97, 119.44, 60.27, 52.20, 48.52, 45.60, 39.52, 30.46, 24.30. HR-MS(TOF): calcd. for C_20_H_23_N_3_O_5_S, [M + H]^+^: 418.1437, found: 418.1431

#### 4.2.12. (2*R*)-1-(4-{4-[(2*R*)-2-Aminopropionamido]phenyl}benzenesulfonyl)pyrrolidine-2-carboxylate Methyl Ester (**7b**)

White solid (134 mg, yield 62.0%) m.p. 157.4~159.1 °C; ^1^H NMR (400 MHz, DMSO-*d*_6_) δ 7.94–7.84 (m, 4H), 7.81 (d, *J* = 9.0 Hz, 2H), 7.74 (d, *J* = 9.0 Hz, 2H), 4.25 (dd, *J* = 8.6, 4.2 Hz, 1H), 3.66 (s, 3H), 3.45 (d, *J* = 6.9 Hz, 2H), 3.20 (d, *J* = 9.8 Hz, 1H), 2.06–1.92 (m, 1H), 1.92–1.76 (m, 2H), 1.63 (d, *J* = 7.0 Hz, 1H), 1.23 (d, *J* = 6.9 Hz, 3H). ^13^C NMR (100 MHz, DMSO-*d*_6_) δ 179.17, 175.28, 172.18, 144.06, 139.59, 135.38, 132.66, 127.90, 127.48, 126.95, 119.52, 60.26, 52.19, 51.18, 48.51, 39.52, 39.52, 30.45, 24.29, 21.47. HR-MS(TOF): calcd. for C_21_H_25_N_3_O_5_S, [M + H]^+^: 432.1593, found: 432.1588.

#### 4.2.13. Methyl (2*R*)-1-(4-{4-[(2*S*)-2-Aminopropionamido]phenyl}benzenesulfonyl)pyrrolidine-2-carboxylate (**7c**)

White solid (141 mg, yield 65.3%), m.p. 155.1~158.3 °C; ^1^H NMR (400 MHz, DMSO-*d*_6_) δ 7.89 (q, *J* = 8.7 Hz, 4H), 7.80 (d, *J* = 8.8 Hz, 2H), 7.74 (d, *J* = 8.9 Hz, 2H), 4.24 (dd, *J* = 8.5, 4.1 Hz, 1H), 3.66 (s, 3H), 3.47 (d, *J* = 6.9 Hz, 1H), 3.21 (dd, *J* = 9.9, 7.2 Hz, 2H), 1.96 (d, *J* = 8.5 Hz, 1H), 1.93–1.78 (m, 2H), 1.63 (d, *J* = 8.2 Hz, 1H), 1.23 (d, *J* = 6.7 Hz, 3H). ^13^C NMR (100 MHz, DMSO-*d*_6_) δ 175.10, 172.19, 147.69, 144.06, 139.57, 135.41, 132.70, 127.91, 127.49, 126.96, 119.54, 60.27, 52.20, 51.11, 48.51, 39.52, 30.45, 24.30, 21.35. HR-MS(TOF): calcd. for C_21_H_25_N_3_O_5_S, [M + H]^+^: 432.1593, found: 432.1588.

#### 4.2.14. Synthesis of (2*R*)-1-{4-[6-[2-(2-Aminoacetamido)pyridin-3-yl]benzenesulfonyl}pyrrolidine-2-carboxylate Methyl Ester (**7d**)

Yellow oil (103 mg, yield 49.1%), m.p. 170.2~173.5 °C; ^1^H NMR (400 MHz, DMSO-*d*_6_) δ 8.75 (s, 1H), 8.24 (s, 2H), 8.04–7.82 (m, 4H), 4.27 (dd, *J* = 8.7, 4.1 Hz, 1H), 3.66 (s, 3H), 3.39 (s, 2H), 3.21 (d, *J* = 8.3 Hz, 2H), 1.97 (d, *J* = 8.1 Hz, 1H), 1.92–1.75 (m, 2H), 1.64 (d, *J* = 6.4 Hz, 1H). ^13^C NMR (100 MHz, DMSO-*d*_6_) δ 172.94, 171.69, 151.53, 146.55, 141.27, 136.92, 135.62, 129.37, 127.98, 127.18, 110.69, 59.59, 50.78, 47.24, 44.24, 30.44, 23.46. HR-MS(TOF): calcd. for C_20_H_24_N_4_SO_5_, [M + H]^+^: 419.1389, found: 419.1384.

#### 4.2.15. Methyl (2*R*)-1-(4-{6-[(2*R*)-2-Aminopropionamido]pyridin-3-yl}benzenesulfonyl)pyrrolidine-2-carboxylate (**7e**)

Yellow oil (96 mg, yield 44.5%), m.p. 146.0~148.4 °C; ^1^H NMR (400 MHz, DMSO-*d*_6_) δ 8.76 (t, *J* = 1.7 Hz, 1H), 8.24 (d, *J* = 2.2 Hz, 2H), 8.01–7.88 (m, 4H), 4.27 (dd, *J* = 8.6, 4.1 Hz, 1H), 3.66 (s, 3H), 3.54 (q, *J* = 7.0 Hz, 1H), 3.41 (ddd, *J* = 9.7, 7.1, 4.9 Hz, 1H), 3.26–3.16 (m, 1H), 2.06–1.93 (m, 1H), 1.87 (d, *J* = 12.4 Hz, 2H), 1.69–1.58 (m, 1H), 1.25 (d, *J* = 7.0 Hz, 3H). ^13^C NMR (100 MHz, DMSO-*d*_6_) δ 175.51, 172.10, 151.59, 146.47, 141.20, 136.81, 136.20, 129.35, 127.93, 127.12, 112.73, 60.24, 52.14, 50.77, 48.44, 39.52, 30.40, 24.24, 20.95. HR-MS(TOF): calcd. for C_20_H_24_N_4_SO_5_, [M + H]^+^: 433.1546, found: 433.1540.

#### 4.2.16. (2*R*)-1-(4-{6-[(2*S*)-2-Aminopropionamido]pyridin-3-yl}benzenesulfonyl)pyrrolidine-2-carboxylate Methyl Ester (**7f**)

Yellow oil (93.5 mg, yield 43.2%), m.p. 146.2~148.9 °C; ^1^H NMR (400 MHz, DMSO-*d*_6_) δ 8.76 (t, *J* = 1.7 Hz, 1H), 8.24 (d, *J* = 1.8 Hz, 2H), 7.99 (d, *J* = 8.5 Hz, 2H), 7.91 (d, *J* = 8.4 Hz, 2H), 4.27 (dd, *J* = 8.6, 4.1 Hz, 1H), 3.67 (s, 3H), 3.55 (q, *J* = 6.9 Hz, 1H), 3.40 (dt, *J* = 9.6, 3.8 Hz, 1H), 3.22 (dt, *J* = 9.6, 7.0 Hz, 1H), 2.07–1.93 (m, 1H), 1.88 (d, *J* = 12.5 Hz, 2H), 1.72–1.55 (m, 1H), 1.25 (d, *J* = 7.0 Hz, 3H). ^13^C NMR (100 MHz, DMSO-*d*_6_) δ 174.28, 170.80, 150.18, 145.58, 139.15, 136.83, 135.71, 129.36, 127.94, 126.51, 112.75, 59.79, 52.14, 50.76, 47.39, 30.40, 24.71, 20.94. HR-MS(TOF): calcd. for C_20_H_24_N_4_SO_5_, [M + H]^+^, 433.1546, found: 433.1540.

#### 4.2.17. (2*S*)-Methyl 1-{4-[4-[4-(2-Aminoacetamido)phenyl]benzenesulfonyl}pyrrolidine-2-carboxylate (**13a**)

White solid (134 mg, yield 64.3%), m.p. 163.4~165.3 °C; ^1^H NMR (400 MHz, DMSO-*d*_6_) δ 7.93–7.85 (m, 4H), 7.80 (d, *J* = 8.9 Hz, 2H), 7.74 (d, *J* = 8.8 Hz, 2H), 4.25 (dd, *J* = 8.6, 4.2 Hz, 1H), 3.66 (s, 3H), 3.40 (dd, *J* = 7.3, 2.4 Hz, 1H), 3.30 (s, 2H), 3.20 (d, *J* = 9.8 Hz, 1H), 1.96 (d, *J* = 6.8 Hz, 1H), 1.87 (d, *J* = 12.0 Hz, 2H), 1.63 (d, *J* = 6.9 Hz, 1H). ^13^C NMR (100 MHz, DMSO-*d*_6_) δ 172.43, 172.20, 144.09, 139.48, 135.39, 132.68, 127.92, 127.56, 126.97, 119.43, 60.27, 52.21, 48.52, 45.65, 39.52, 30.46, 24.31. HR-MS(TOF): calcd. for C_20_H_23_N_3_O_5_S, [M + H]^+^: 418.1437, found: 418.1431.

#### 4.2.18. (2*S*)-1-(4-{4-[(2*R*)-2-Aminopropionamido]phenyl}benzenesulfonyl)pyrrolidine-2-carboxylate Methyl Ester (**13b**)

White solid (136 mg, yield 63.3%), m.p. 156.4~159.1 °C. ^1^H NMR (400 MHz, DMSO-*d*_6_) δ 7.93–7.85 (m, 4H), 7.81 (d, *J* = 8.8 Hz, 2H), 7.74 (d, *J* = 8.8 Hz, 2H), 4.25 (dd, *J* = 8.6, 4.2 Hz, 1H), 3.66 (s, 3H), 3.45 (d, *J* = 6.9 Hz, 2H), 3.21 (d, *J* = 9.8 Hz, 1H), 1.98 (s, 1H), 1.87 (d, *J* = 11.9 Hz, 2H), 1.64 (d, *J* = 7.0 Hz, 1H), 1.23 (d, *J* = 6.9 Hz, 3H). ^13^C NMR (100 MHz, DMSO-*d*_6_) δ 175.29, 172.19, 144.06, 139.59, 135.38, 132.66, 127.90, 127.48, 126.95, 119.52, 60.26, 52.20, 51.18, 48.51, 39.52, 30.45, 24.30, 21.47. HR-MS(TOF): calcd. for C_21_H_25_N_3_O_5_S, [M + H]^+^: 432.1593, found: 432.1588.

#### 4.2.19. Methyl (2*S*)-1-(4-{4-[(2S)-2-Aminopropionamido]phenyl}benzenesulfonyl)pyrrolidine-2-carboxylate (**13c**)

White solid (141.7 mg, yield 65.6%), m.p. 155.3~157.6 °C; ^1^H NMR (400 MHz, DMSO-*d*_6_) δ 7.94–7.85 (m, 4H), 7.81 (d, *J* = 8.8 Hz, 2H), 7.74 (d, *J* = 9.0 Hz, 2H), 4.25 (dd, *J* = 8.6, 4.2 Hz, 1H), 3.66 (s, 3H), 3.46 (d, *J* = 6.9 Hz, 2H), 3.20 (d, *J* = 9.8 Hz, 1H), 2.08–1.92 (m, 1H), 1.93–1.76 (m, 2H), 1.63 (dd, *J* = 7.0, 5.0 Hz, 1H), 1.23 (d, *J* = 6.9 Hz, 3H). ^13^C NMR (100 MHz, DMSO-*d*_6_) δ 175.29, 172.19, 144.07, 139.60, 135.39, 132.66, 127.91, 127.48, 126.95, 119.52, 60.26, 52.19, 51.18, 48.51, 39.52, 30.44, 24.29, 21.47. HR-MS(TOF): calcd. for C_21_H_25_N_3_O_5_S, [M + H]^+^: 432.1593, found: 432.1588.

#### 4.2.20. (2*S*)-1-{4-[6-[2-(2-Aminoacetamido)pyridin-3-yl]benzenesulfonyl}pyrrolidine-2-carboxylate Methyl Ester (**13d**)

Yellow oil (109 mg, yield 52.1%), m.p. 170.3~172.9 °C; ^1^H NMR (400 MHz, DMSO-*d*_6_) δ 8.80 (dd, *J* = 2.5, 0.9 Hz, 1H), 8.28 (dd, *J* = 8.7, 2.5 Hz, 1H), 8.18 (d, *J* = 8.8 Hz, 1H), 8.00 (d, *J* = 8.6 Hz, 2H), 7.92 (d, *J* = 8.5 Hz, 2H), 4.27 (dd, *J* = 8.6, 4.1 Hz, 1H), 3.85 (s, 2H), 3.66 (s, 3H), 3.25–3.17 (m, 1H), 2.04–1.92 (m, 1H), 1.92–1.77 (m, 2H), 1.64 (dt, *J* = 7.1, 5.2 Hz, 1H), 1.22 (d, *J* = 3.0 Hz, 1H). ^13^C NMR (100 MHz, DMSO-*d*_6_) δ 172.15, 167.00, 151.34, 146.63, 141.10, 137.07, 136.30, 129.77, 127.99, 127.27, 113.30, 60.28, 52.21, 48.49, 39.52, 30.45, 24.30. HR-MS(TOF): calcd. for C_19_H_22_N_4_O_5_S, [M + H]^+^: 419.1389, found: 419.1384.

#### 4.2.21. Methyl (2*S*)-1-(4-{6-[(2*R*)-2-Aminopropionamido]pyridin-3-yl}benzenesulfonyl)pyrrolidine-2-carboxylate (**13e**)

Yellow oil (103.7 mg, yield 47.9%), m.p. 146.2~148.9 °C; ^1^H NMR (400 MHz, CDCl_3_-*d*) δ 8.61–8.53 (m, 1H), 8.36 (d, *J* = 8.7 Hz, 1H), 7.96 (d, *J* = 8.4 Hz, 3H), 7.70 (d, *J* = 8.4 Hz, 2H), 4.37 (dd, *J* = 8.2, 3.9 Hz, 1H), 3.73 (s, 3H), 3.56–3.48 (m, 1H), 3.36 (d, *J* = 9.5 Hz, 1H), 2.14–1.93 (m, 4H), 1.89–1.76 (m, 1H), 1.24 (s, 3H). ^13^C NMR (100 MHz, CDCl_3_-*d*) δ 174.78, 172.66, 151.35, 146.50, 142.09, 137.47, 137.01, 130.97, 128.39, 127.30, 113.72, 77.16, 60.54, 52.62, 51.39, 48.53, 31.08, 29.82, 24.84. HR-MS(TOF): calcd. for C_20_H_24_N_4_SO_5_, [M + H]^+^: 433.1546, found: 433.1540.

#### 4.2.22. (2*S*)-1-(4-{6-[(2*S*)-2-Aminopropionamido]pyridin-3-yl}benzenesulfonyl)pyrrolidine-2-carboxylate Methyl Ester (**13f**)

Yellow oil (98 mg, yield 45.3%), m.p. 146.8~149.1 °C; ^1^H NMR (400 MHz, CDCl_3_-*d*) δ 8.56 (s, 1H), 8.35 (d, *J* = 8.5 Hz, 1H), 7.96 (d, *J* = 8.4 Hz, 3H), 7.70 (d, *J* = 8.2 Hz, 2H), 4.37 (dd, *J* = 8.2, 3.9 Hz, 1H), 3.73 (s, 3H), 3.51 (d, *J* = 7.1 Hz, 1H), 3.38 (t, *J* = 7.9 Hz, 1H), 2.10–1.96 (m, 4H), 1.88–1.77 (m, 1H), 1.25 (s, 3H). ^13^C NMR (100 MHz, CDCl_3_-*d*) δ 174.78, 172.66, 151.35, 146.50, 142.09, 137.47, 137.01, 130.97, 128.39, 127.30, 113.72, 77.16, 60.54, 52.62, 51.39, 48.53, 31.08, 29.82, 24.84. HR-MS(TOF): calcd. for C_20_H_24_N_4_SO_5_, [M + H]^+^: 433.1546, found: 433.1510.

### 4.3. In Vitro Antiproliferative Activities and Cytotoxicity Studies

RPMI 8226 and HL-7702 cells were cultured in RPMI-1640 medium, Hela cells were cultured in MEM medium, and HL-60 cells were cultured in IMDM medium with 1% penicillin–streptomycin solution and 10% FBS, respectively. The above cells were cultured at 37 °C and 5% CO_2_. After 24 h of cell spreading, 99 μL of medium per well was prepared and 1 μL of the compound prepared solution was added to the wells. Then, the solution was incubated at 37 °C in a 5% CO_2_ incubator for 72 h. The cell plates to be tested were left at room temperature and 100 μL of medium per well was discarded. Then, 100 μL of CTG reagent was added, placed in a rapid shaker for 2 min, and left at room temperature away from light for 30 min. The chemiluminescence signal was read by an envision multifunctional enzyme marker. The inhibition ratios and IC_50_ values were calculated using Prism Graph Pad software.

### 4.4. Crystallography

Single crystal X-ray diffraction data of **2** and **8** were collected by CrysAlisPro [48] 1.171.39.46e Agilent Technologies, on a Xcalibur, Atlas, Gemini ultra diffractometer at 293 K under the Cu Kα radiation. Compounds **7a**, **7d**, and **13a** were collected by Bruker APEX-II CCD diffractometer at 150 K, 302 K, and 170 K, respectively, under Cu Kα radiation. The structure solutions of **2** and **8** were prepared using SHELXS (Sheldrick, 2008) [49] and refined by SHELXL 2018/3 (Sheldrick, 2015) [50]. The structure solutions of **7a**, **7d**, and **13a** were prepared using SHELXT 2018/2 (Sheldrick, 2015) [51] and refined by SHELXL 2018/3. The C-H hydrogen atoms were geometrically positioned and treated as riding atoms where C–H = 0.93 Å with Uiso(H) = 1.2 Ueq(C) for aromatic carbon atoms and C–H = 0.96 Å with Uiso(H) = 1.5 Ueq(C) for methyl carbon atoms.

### 4.5. Reverse Docking

#### 4.5.1. Protein Preparation

The crystal structures of different HDACs isoforms were obtained from the Protein Data Bank (https://www.rcsb.org/ (accessed on 2 April 2022)). HDAC1 (5ICN), HDAC2 (4LXZ), HDAC3 (4A69), HDAC4 (2VQM), HDAC6 (5EDU), HDAC7 (3C10), HDAC 8 (1T69), and HDAC 10 (6WDY) were selected as the docking targets. All these structures are human protein constructs, except HDAC10 (6WDY), which is a zebra fish (Danio rerio) construct. Their X-ray resolutions are listed in Table 1. The crystal structures have not been reported for HDAC5, HDAC9, and HDAC11. Therefore, we obtained a 3D protein model for HDAC5 (Q9UQL6) and HDAC9 (Q9UKV0) using SWISS-MODEL [52] (https://swissmodel.expasy.org/ (accessed on 2 April 2022)). The 3D protein model for HDAC 11 (Q96DB2) did not possess a Zn^2+^ ion; thus, HDAC11 was not selected as a docking target.

After importing the protein target into the Schrödinger software, the structure of the multiple chained protein was preprocessed to a single unit, and the unwanted ligands, water molecules, K^+^ ions, etc. were eliminated outside the binding pocket. Then, these protein structures were subjected to protein preparation wizard, where residue bond orders were fixed, missing hydrogens were added, zero-order bonds to metals were created, disulphide bonds were created, and het states were generated using Epik at pH: 7.0 ± 2.0. Finally, preprocessed protein was optimized with PROPKA and then minimized with the OPSL4 force field [53], followed by a convergence of heavy atoms of RMSD 0.3 Å.

#### 4.5.2. Generation of Receptor Grid

After protein preparation, the receptor grid for HDAC isoforms was generated at the centroid of the co-crystal ligand using receptor grid generation program. The co-crystal ligand of HDAC3 is acetic acid molecule, which is too small to generate the docking box. Therefore, the docking box was generated by selecting the key amino acid residues, including HID 134, HID 135, GLY 143, PHE 144, HID 172, PHE 200, and TYR 298. Due to the lack of ligands for the SWISS-MODEL of HDAC5 and HDAC9, the docking box was generated in the same way as for HDAC3. The HDAC 5 protein grid box was generated, including HID 747, HID 748, PHE 757, HID 787, PHE 816, ASP 879, and LEU 888 residues. The HDAC9 grid box was made by including HID 782, HID 783, PHE 792, ASP 820, HID 822, PHE 851, and HID956. Finally, the Zn^2+^ ion was treated as a constraint atom that could form metal–ligand interactions during docking, and the coordination geometry of the Zn^2+^ ion was set as tetrahedral or octahedral, which was predicted by the Zn^2+^ ion environment in the receptor.

#### 4.5.3. Ligand Preparation

The structures of ligands were prepared using ChemDraw Ultra and Chem 3D software. The ligands were prepared using the Ligprep module, where they were desalted after the addition of hydrogen atoms, followed by the generation of all ionization states possible at the physiological pH:7.0 ± 2.0. Epik was used for this purpose, and ‘Add metal binding states’ was selected. In the stereoisomer generation setting, ‘determine chiralities from the 3D structure’ were selected.

#### 4.5.4. Docking and Reverse Docking

The prepared receptor grid and ligand were subject to the docking module (Glide [54,55]) and docked using standard precision (SP) docking methods. Constrains to metal and coordination geometry were used in docking. Each ligand was set to write out ten docked conformations at most. The optimal docking conformation was subject to the superposition module to calculate RMSD. Reverse docking was performed via the virtual screening workflow module, where the prepared ligand was inputted and used directly for subjobs without any preparation and 10 receptor grids were inputted at once. Epik state penalties for docking and Glide SP docking methods were used. Up to 10 poses per compound state was set for generation purposes.

#### 4.5.5. Molecular Dynamics

Molecular dynamics (MDs) simulations were performed using the desmond program. The neutral territory method (midpoint method) was adopted to efficiently exploit a high degree of computational parallelism. The OPLS4 force-field model was used to analyze amino acid interactions in protein and the TIP3P method was used for the water model. The equilibration of the system was passed out using the default protocol provided in Desmond, which consists of a series of restrained minimizations and molecular dynamics simulations that are designed to slowly relax the system without deviating substantially from the initial protein coordinates. The TIP3P water molecules were added. The orthorhombic dimensions of each water box were 10 Å × 10 Å × 10 Å approximately, which confirmed that the whole surfaces of the complexes ought to be covered. The neutralization of system was carried out by adding Cl counter ions to balance the net charge of the system. After the construction of the solvent environment, each complex system was composed of about 91,372 atoms. Before equilibration and the long production MD simulations, the systems were minimized and pre-equilibrated using the default relaxation routine implemented in Desmond. The whole system was subjected to 300 K for 100 ns of simulation of the protein–ligand complex. RMSD plots, RMSF plots, ligand interaction diagrams, histogram plots, etc., were generated through simulation interactions of the diagram module.

## Data Availability

All data are contained within the article or supplementary material. The numerical data represented in the figures are available upon request from the corresponding author.

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
