# Peer review of "Design, Synthesis, Bioactivity Evaluation, Crystal Structures, and In Silico Studies of New α-Amino Amide Derivatives as Potential Histone Deacetylase 6 Inhibitors"

_molecules, 2022, doi:10.3390/molecules27103335_

Round 1
Reviewer 1 Report
Xu et al submitted title, "Design, Synthesis, Bioactivity Evaluation, Crystal Structures, DFT calculation and Molecular docking of New Stereoisomeric
α-Amino Amide Based Non-Hydroxamate HDAC6 Inhibitors" is organized in a systemic way. However, there are some points which need to be revised by the authors.
1. Figure 2. As it contains surface mapping, therefore please include information in the figure caption, related to the meaning of these colors (blue, red, green) which are representing the cavity.
2. As authors doesn't perform HDAC/HDAC-6 protein binding/affinity assays, but performed molecular modelling, therefore there are concerns related to the validation of the study :
a. Self docking: In order to verify the correctness of docking method, self docking is mainly performed. In self docking, the co-crystal ligand of PDB (TRICHOSTATIN A) is extracted or independent build, which later then docked into its PDB. The deviation (RMSD) between the actual binding conformation of ligand in its co-crystal structure can be then compared with its docked conformation (doi.org/10.1016/j.jsps.2018.01.017). Please perform self docking and mention the RMSD.
b. Author must explain/justify how they find out the compounds as selective HDAC-6 inhibitors but not as non-selective HDAC inhibitors. One possible way to minimize such scenario is perform inverse docking where HDAC isoforms (1-9, 11) can docked with these compounds (10.1016/j.jmgm.2010.09.004; doi.org/10.1016/j.ejmech.2019.04.064).
b. Generally researcher prefer either, "molecular docking with molecular dynamics on the best docking pose" or "induced/flexible-docking". While, authors used DFT calculations followed by molecular docking. However, author tried to explain the results, but those are in more general way.
"Comparing 7a with 7d, the α-amino amide in 7d is rotated about 180° relative to 7a, causing a large difference in the electrostatic potential on the molecular surface. Not only does this affect the interaction of the molecule with the protein cavity, but it may also affect the ability of α-amino amides to chelate with zinc ions"
Keeping in mind for the audience of a wide background, authors could connect this point and utilize it to support their findings observed in the next experiment (i.e docking).
3. There should a justification provided by authors about their selection of cell lines or why not other. Were there any HDAC dependent cell lines tested, that are reported in the paper?
4. There are some typographical errors, it is suggested for authors to read the manuscript carefully.
The manuscript illustrates a broad range of experiments to investigate the role of α-Amino Amide Based Non-Hydroxamate as cytotoxic and putative HDAC inhibitors, and therefore the manuscript certainly has merit. However, the lack of sufficient justification of how the reported compounds can be selective HDAC-6 inhibitor, needs to be addressed along with other points raised in above comments before considering in the current journal.
Reviewer 2 Report
The paper submitted by Junhai Xiao and colleagues includes a very comprehensive research work, that goes from in silico studies to bioactivity evaluation, of HDAC6 Inhibitors. The topic is interesting for medicinal chemists as HDAC is a relevant drug target, and the paper brings a strong contribution from the synthetic and structural point of view. On the other hand, some concerns arise from the study of bioactivity and molecular docking.
- About biological activity, do the authors have proof on the mechanism of the molecules on the enzyme? In fact, according to the paper, this is a proposed mechanism of action, since only antiproliferative activity is indeed tested.
- Molecular docking of ligands in pockets containing metals can be challenging. Nevertheless, from docking poses (Figures 7-9), it looks like the orientation of the functional group is not correct to chelate Zn. Authors should check and comment on this point. Moreover, more details about docking procedures must be explained in the Materials and Methods section.
Additionally, I report some minor comments below:
- Abstract and Introduction are well written and informative. Anyway, I suggest avoiding the use of HDAC6 abbreviation without explaining its meaning in title and abstract.
- Figure 2: authors should improve figure quality. In particular, concerning panel C, residue labels are rather small and difficult to read. More importantly, from what observed in panel A, it looks like the molecule and Zn are superimposing. Please check.
Round 2
Reviewer 1 Report
The authors made various changes as were asked in the previous draft, therefore the manuscript has enough elements to be considered in the current journal.
Author Response
Thank you very much for your comments.
Reviewer 2 Report
The authors explained in their answer how they performed the selection of a certain software instead of another. I would only suggest to provide a clearer definition of reverse docking.
Author Response
Thank you very much for your comments. We have explained in detail about reverse molecular docking according to your request, please refer to page 4, lines 99-105.
Virtual screening has become an important part of the drug discovery process. Molecular docking-based virtual screening involves docking and scoring a library of small molecule compounds against a given target protein, from which potential ligand molecules for the target protein are screened. Reverse molecular docking is the opposite of molecular docking in that it involves docking and scoring an active molecule with multiple or a large number of protein targets, and thus predicting the potential targets of the active molecule.